# Dietary exposure to nitrites and nitrates in association with type 2 diabetes risk: Results from the NutriNet-Santé population-based cohort study

Bernard Srour[1,2]*, Eloi Chazelas[1,2], Nathalie Druesne-Pecollo[1,2], Younes Esseddik[1], Fabien Szabo de Edelenyi[1], Cédric Agaësse[1], Alexandre De Sa[1], Rebecca Lutchia[1], Charlotte Debras[1,2], Laury Sellem[1,2], Inge Huybrechts[2,3], Chantal Julia[1,4], Emmanuelle Kesse-Guyot[1,2], Benjamin Allès[1], Pilar Galan[1,2], Serge Hercberg[1,2,4], Fabrice Pierre[2,5], Mélanie Deschasaux-Tanguy[1,2], Mathilde Touvier[1,2]

1 Sorbonne Paris Nord University, Inserm U1153, Inrae U1125, Cnam, Nutritional Epidemiology Research Team (EREN), Epidemiology and Statistics Research Center–University of Paris-Cité (CRESS), Bobigny, France, 2 Nutrition And Cancer Research Network (NACRe Network), Jouy-en-Josas, France, 3 International Agency for Research on Cancer, World Health Organization, Lyon, France, 4 Public Health Department, Avicenne Hospital, AP-HP, Bobigny, France, 5 Toxalim (Research Centre in Food Toxicology), Université de Toulouse, INRAE, ENVT, INP-Purpan, UPS, Toulouse, France

* b.srour@eren.smbh.univ-paris13.fr

**Data Availability Statement:** Data described in the manuscript, code book, and analytic code will be made available upon request pending application

## Abstract

### Background

Nitrites and nitrates occur naturally in water and soil and are commonly ingested from drinking water and dietary sources. They are also used as food additives, mainly in processed meats, to increase shelf life and to avoid bacterial growth. Experimental studies suggested both benefits and harmful effects of nitrites and nitrates exposure on type 2 diabetes (T2D) onset, but epidemiological and clinical data are lacking. We aimed to study these associations in a large population-based prospective cohort study, distinguishing foods and water-originated nitrites/nitrates from those from food additives.

### Methods and findings

Overall, 104,168 adults from the French NutriNet-Santé cohort study (2009 to 2021, 79.1% female, mean age [SD] = 42.7 [14.5]) were included. Associations between self-reported exposure to nitrites and nitrates (evaluated using repeated 24-h dietary records, linked to a comprehensive food composition database and accounting for commercial names/brands details of industrial products) and risk of T2D were assessed using cause-specific multivariable Cox proportional hazard models adjusted for known risk factors (sociodemographic, anthropometric, lifestyle, medical history, and nutritional factors). During a median follow-up duration of 7.3 years (interquartile range: [3.2; 10.1] years), 969 incident T2D cases were ascertained. Total nitrites and foods and water-originated nitrites were both positively associated with a higher T2D risk (HR$_{tertile\ 3\ vs.1}$ = 1.27 (95% CI 1.04 to 1.54), P$_{trend}$ = 0.009 and 1.26 (95% CI 1.03 to 1.54), P$_{trend}$ = 0.02, respectively). Participants with higher exposure to

and approval. Researchers from public institutions can submit a collaboration request including information on the institution and a brief description of the project to collaboration@etude-nutrinet-sante.fr. All requests will be reviewed by the steering committee of the NutriNet-Santé study. If the collaboration request is accepted, a data access agreement will be necessary and appropriate authorizations from the competent administrative authorities may be needed. In accordance with existing regulations, no personal identification data will be accessible.

**Funding:** The NutriNet-Santé study was supported by the following public institutions: Ministère de la Santé, Santé publique France, Institut National de la Santé et de la Recherche Médicale (INSERM), Institut National de la Recherche Agronomique (INRAE), Conservatoire National des Arts et Métiers (CNAM) and University Sorbonne Paris Nord. EC was supported by a Doctoral Funding from University Sorbonne Paris Nord - Galilée Doctoral School. CD was supported by a grant from the French National Cancer Institute (INCa). This project (MT) has received funding from the European Research Council (ERC) under the European Union's Horizon 2020 research and innovation program (grant agreement No 864219), the French National Cancer Institute (INCa_14059), the French Ministry of Health (arrêté 29.11.19), the IdEx Université de Paris (ANR-18-IDEX-0001) and a Bettencourt-Schueller Foundation Research Prize 2021. This project was awarded the NACRe (French network for Nutrition And Cancer Research) Partnership Label. This work only reflects the authors' view and the funders are not responsible for any use that may be made of the information it contains. Researchers were independent from funders. The funders had no role in study design, data collection and analysis, decision to publish, or preparation of the manuscript.

**Competing interests:** I have read the journal's policy and the authors of this manuscript have the following competing interests: FP received funding from the IFIP (French Pork Institute), for another experimental project on processed meats and colorectal cancer, aiming to evaluate technical solutions (e.g. agricultural practices, formulation) to mitigate the well-established deleterious impact of processed meat on colorectal cancer risk. It is not related at all to the present project/manuscript nor to type-2 diabetes. IFIP had no role in designing the present study nor funding it. All other authors declare that they have no conflict of interest.

additives-originated nitrites (i.e., above the sex-specific median) and specifically those having higher exposure to sodium nitrite (e250) had a higher T2D risk compared with those who were not exposed to additives-originated nitrites (HR $_{higher\ consumers\ vs.\ non-consumers}$ = 1.53 (95% CI 1.24 to 1.88), $P_{trend}$ < 0.001 and 1.54 (95% CI 1.26 to 1.90), $P_{trend}$ < 0.001, respectively). There was no evidence for an association between total, foods and water-originated, or additives-originated nitrates and T2D risk (all $P_{trend}$ = 0.7). No causal link can be established from this observational study. Main limitations include possible exposure measurement errors and the lack of validation versus specific nitrites/nitrates biomarkers; potential selection bias linked to the healthier behaviors of the cohort's participants compared to the general population; potential residual confounding linked to the observational design, as well as a self-reported, yet cross-checked, case ascertainment.

## Conclusions

The findings of this large prospective cohort did not support any potential benefits for dietary nitrites and nitrates. They suggested that a higher exposure to both foods and water-originated and additives-originated nitrites was associated with higher T2D risk in the NutriNet-Santé cohort. This study provides a new piece of evidence in the context of current debates about updating regulations to limit the use of nitrites as food additives. The results need to be replicated in other populations.

## Trial registration

ClinicalTrials.gov NCT03335644 (https://clinicaltrials.gov/ct2/show/NCT03335644)

## Author summary

### Why was this study done?

- In several countries, debates recently emerged regarding a potential banning of nitrites and nitrates as food additives.

- While experimental studies are accumulating and seem to support a prohibiting strategy, epidemiological evidence regarding type 2 diabetes (T2D) risk remains unfortunately limited, yet seems important to develop given the suggested role of N-nitroso compounds in the development of insulin resistance.

- Only 1 small prospective cohort study in Iran ($n$ = 2,193) observed direct associations between higher dietary nitrite intakes and a higher T2D risk, in participants with low vitamin C intakes. However, this study did not distinguish between foods and water-originated nitrites and food additives-originated nitrites, and did not collect data on specific consumed brands.

### What did the researchers do and find?

- We used appropriate statistical models to investigate, in the large prospective cohort NutriNet-Santé ($n$ = 104,168), whether dietary exposure to nitrites/nitrates was

**Abbreviations:** BW, body weight; EFSA, European Food Safety Authority; eNOS, endothelial NO synthase; GNPD, Global New Products Database; GSFA, General Standard for Food Additives; HOMA, homeostatic model assessment; IARC, International Agency for Research on Cancer; IGF, insulin-like growth factor; IPAQ, International Physical Activity Questionnaire; NO, nitric oxide; NOC, N-nitroso compound; SSB, sugar-sweetened beverage; T2D, type 2 diabetes.

associated with T2D risk, while differentiating foods and water-originated nitrites/nitrates from those from food additives.

- We found for the first time to our knowledge, an association between additives-originated nitrites and specifically sodium nitrite (e250), with T2D risk ($HR_{\text{higher consumers vs. non-consumers}} = 1.54$ (95% CI 1.26 to 1.90), $P_{\text{trend}} < 0.001$).

- We also found associations between exposure to total dietary nitrites and a higher T2D risk ($HR_{\text{tertile 3 vs.1}} = 1.27$ (95% CI 1.04 to 1.54), $P_{\text{trend}} = 0.009$).

### What do these findings mean?

- Our findings suggest a direct association between additives-originated nitrites and T2D risk, and corroborate previously suggested associations between total dietary nitrites and T2D risk.

- The main limitations of the study include possible exposure assessment errors, potential selection bias linked to the healthier behaviors of the cohort's participants compared to the general population, and potential residual confounding linked to the observational design.

- Although they need confirmation by other prospective studies and experimental research, these results provide a new piece of evidence in the context of current discussions regarding the reduction of additives-originated nitrites in the food industry and could support the need for better regulation of soil contamination.

## Introduction

Nitrites and nitrates are commonly ingested from drinking water and miscellaneous dietary sources, as they are naturally present in water and soil [1]. The use of nitrogen fertilizers during agricultural practices is responsible for groundwater and surface water pollution by nitrates as well as soil health degradation, which is a rising problem for both human health and ecosystems [2]. The main sources of foods-originated nitrites and nitrates are green leafy vegetables and beetroots [3]. Dietary exposure to nitrites and nitrates also includes food additives, as they can be used as preservatives to improve shelf life, also providing a pink coloration to ham and several processed meats [1]. Nitrites and nitrates are massively used as food additives; more than 15,000 packaged items on the French market currently contain added nitrites or nitrates [4].

Apart from dietary exposure to nitrates and nitrites, the nitrate anion ($NO_3^-$) is generated endogenously, and is involved in several signaling pathways [5]. About 25% of the ingested exogenous nitrate is reduced to nitrite by commensal bacteria residing in the mouth, of which about 20% is converted to nitric oxide (NO) in the stomach, through the nitrate-nitrite-NO pathway (i.e., about 5% of ingested nitrate is converted to NO), providing a continuous source of NO for human body, in addition to NO production through the L-arginine oxidative pathway [5,6]. Mounting evidence led the International Agency for Research on Cancer (IARC) to classify ingested nitrate or nitrite under conditions that result in endogenous nitrosation, leading to the formation of N-nitroso compounds (NOCs), among which nitrosamines, as

probably carcinogenic to humans (group 2A) [1], and as proven carcinogens in a number of animal species [7,8]. In contrast, their metabolic impact is unclear, since experimental in vitro/ in vivo studies described either potentially beneficial or deleterious metabolic effects of dietary nitrates and nitrites. For example, in mice lacking endothelial NO synthase (eNOS) enzymes (considered as the main source of endogenous NO production, through the L-arginine oxidative pathway) at high risk of metabolic syndrome-like phenotype, nitrate supplementation, allowing the endogenous production of NO by the sequential reduction of nitrate to nitrite and then NO, through the alternative nitrate-nitrite-NO pathway, reduced body weight (BW), improved glucose tolerance, and decreased plasma triglycerides and levels of glycated hemoglobin (HbA1c) [9]. Another study suggested that vegetable-derived nitrate in mice fed a high-fat and high-fructose diet improved insulin sensitivity in the homeostatic model assessment (HOMA) [10]. On the other hand, some experimental studies suggested that the formation of nitrosamines might have adverse effects on insulin/insulin-like growth factor (IGF) signaling pathways and pancreatic β cell functions [11,12], therefore potentially playing a role in insulin resistance development and its related metabolic disorders.

However, there is a lack of human data regarding the role of dietary nitrites and nitrates in metabolic dysfunction and type 2 diabetes (T2D) onset [5]. Indeed, to our knowledge, only 1 small prospective cohort study in Iran ($n$ = 2,193) explored and found direct associations between higher dietary nitrite intakes, assessed using a 168-item semi-quantitative food frequency questionnaire, and a higher T2D risk, in participants who had a low vitamin C intake (<108 mg/d) [13]. In this study, no difference was made between foods and water sources of nitrites/nitrates and food additive sources, even though animal/plant-based sources were used as a proxy for the source, and the specific exposure to nitrite/nitrate additives (potassium nitrite (European code e249), sodium nitrite (e250), sodium nitrate (e251), and potassium nitrate (e252)) was not evaluated.

In a context where several public health authorities around the world are questioning a possible suspension of the use of nitrites and nitrates as food additives [14], we aimed to investigate, in the large prospective cohort NutriNet-Santé, whether dietary exposure to nitrites/ nitrates was associated with T2D risk, while differentiating those originated from food and water, from those originated from food additives.

## Material and methods

### Study population

The NutriNet-Santé study aims to investigate the associations between nutrition and health. It is an ongoing web-based cohort launched in May 2009 in France. Participants aged 15 years or above with access to the internet are continuously recruited since May 2009 among the general population. Details have been reported elsewhere [15]. Enrolled participants completed questionnaires using a dedicated online platform (etude-nutrinet-sante.fr). The NutriNet-Santé study is conducted according to the Declaration of Helsinki guidelines and was approved by the Institutional Review Board of the French Institute for Health and Medical Research (IRB Inserm n˚0000388FWA00005831) and the "Commission Nationale de l'Informatique et des Libertés" (CNIL n˚908450/n˚909216). It is registered at clinicaltrials.gov as NCT03335644. Electronic informed consent is obtained from each participant. The NutriNet-Santé study was developed to investigate the relationships between multiple dietary exposures and the incidence of chronic diseases, such as T2D. The general protocol of the cohort, written in 2008 before the beginning of the study, is available online [16]. Regarding food additives specifically, the present work is part of a series of prespecified analyses that are included in a project funded

by the European Research Council (https://erc.europa.eu/news-events/magazine/erc-2019-consolidator-grants-examples#ADDITIVES).

## Data collection

At baseline, participants completed a batch of 5 questionnaires related to sociodemographic and lifestyle characteristics (e.g., sex, date of birth, occupation, educational level, smoking status), anthropometry (e.g., height, weight) [17], physical activity (7-day International Physical Activity Questionnaire [IPAQ]) [18], health status, and dietary intakes. Participants were invited to complete a series of 3 non-consecutive validated web-based 24-h dietary records at baseline and every 6 months, randomly assigned over a 2-week period (2 weekdays and 1 weekend day). In this prospective study, we averaged dietary intakes from all 24-h dietary records available during the first 2 years of each participant's follow-up, considering them as baseline usual dietary intakes. The NutriNet-Santé web-based self-administered 24-h dietary records have shown good performances when tested against an interview by a trained dietitian [19] and against blood and urinary biomarkers (showing reasonable estimates of true intakes of fruits, vegetables, fish, beta carotene, vitamin C, n-3 fatty acids, proteins, and potassium) [20,21]. The dietary assessments included details of commercial names/brand of industrial products, to properly estimate individual additive exposure. All foods and beverages consumed during a 24-h period for each of the 3 main meals (breakfast, lunch, and dinner) and any other eating occasion were declared using the platform, with an estimation of portion sizes based on validated photographs or quantified weight/volume. Dietary underreporting was identified on the basis of the method proposed by Black [22], using the basal metabolic rate and Goldberg cutoff, and under-energy reporters were excluded (details and comparison between under-reports and included participants in Method A in S1 Appendix). Mean daily alcohol, micro- and macro-nutrient, and energy intakes were calculated using the NutriNet-Santé food composition database, containing more than 3,500 different items [23]. In addition, fasting blood samples were collected for 19,772 participants.

## Exposure to nitrites and nitrates

The detailed assessment method of nitrites and nitrates has been described elsewhere [24]. Briefly, we estimated intakes of nitrites/nitrates as (1) foods and water-originated (non-additive sources), i.e., naturally occurring in food products and water contamination; and (2) food additives-originated (detailed in Method B in S1 Appendix), that is exposure to potassium nitrite (e249), sodium nitrite (e250), sodium nitrate (e251), and potassium nitrate (e252). We also determined the exposure to total nitrites/nitrates, reflecting the exposure from both foods, water, and food additives.

Foods-originated nitrites/nitrates were determined by food category using the European Food Safety Authority (EFSA)'s concentration levels for natural sources and contamination from agricultural practices [25,26]. The French official regional sanitary control of tap water was used to estimate intakes via water consumption, by region of residence [27]: SISE-Eaux is a French governmental database including data on quality control of tap water in the 34,955 French municipalities. It is coordinated by the Regional Health Agencies and consists of a collection of 310,000 samples per year (12 million samples overall). Data are available publicly [27] and a municipality-specific merge according to the NutriNet-Santé participants' postal code has been performed, as well as a dynamic temporal merge according to the year of dietary records. As regards all food additives in the NutriNet-Santé cohort, a double qualitative/quantitative approach was used: The presence of food additives was determined using 3 databases: OQALI [28], a national database hosted by the French food safety authority (ANSES) and

National Research Institute for Agriculture, Food and the Environment (INRAE) to characterize the quality of the food supply, Open Food Facts [4], an open collaborative database of food products marketed worldwide, and Mintel Global New Products Database (GNPD) [29], an online database of innovative food products in the world. Quantitative assessment of food additives was performed, using in that order (1) ad hoc laboratory assays; (2) doses of generic foods as reported by EFSA; and (3) from the Codex General Standard for Food Additives (GSFA) [30] in case the first 2 options were not available. The detailed decision tree is described in Method B in S1 Appendix. Despite the validation of dietary records against blood and urinary markers for energy and key nutrients, specific exposure to nitrates and nitrates has not been validated against blood or urine assays, given the challenge in identifying specific biomarkers reflecting exogenous dietary exposure and not endogenous metabolism. Therefore, no objective information on the validity, sensitivity, or specificity of the exposure assessment was available.

## Case ascertainment

Participants were asked to declare major health events though the yearly health questionnaire, through a specific health check-up questionnaire every 6 months, or at any time through a specific interface on the study website. They were also asked to declare all medications they use via the check-up and yearly questionnaires. Besides, data from NutriNet-Santé are linked to medico-administrative databases of the national health insurance (SNIIRAM) database (Decree in the Council of State (n°2013–175)), providing detailed information about the reimbursement of medication and medical consultations. T2D cases were ascertained using a multisource approach, i.e., T2D declaration during follow-up along with declaration of the use of T2D medication (or a reimbursement of T2D medication detected from SNIIRAM) or hyperglycemia in the biological data along with 1 T2D medication use during follow-up. The detailed T2D case ascertainment is presented in Method C in S1 Appendix. All incident T2D cases ascertained up to October 1, 2021 were considered in this study.

## Statistical analyses

A total of 104,168 participants reporting no prevalent T2D at baseline and who provided at least 2 valid 24-h dietary records during their first 2 years of follow-up were included. Flowchart is presented in Figure A in S1 Appendix. We defined sex-specific tertiles of intakes of nitrites and nitrates (total and foods and water-originated), based on the whole population. For additives-originated nitrites and nitrates, as the number of non-consumers was substantial (>25%), 3 categories of intakes were defined: non-consumers, low consumers, and high consumers (the latter 2 being separated by sex-specific median among consumers). Potassium nitrite (e249) and sodium nitrate (e251) were consumed by less than 1% of the population. They were therefore considered in the total food additive analyses, but their individual associations with T2D risk were not studied. Cause-specific Cox proportional hazards models with age as the primary timescale were used to study the associations between the exposure to nitrites and nitrates with T2D risk, with death and type 1 diabetes handled as competing events (i.e., an event whose occurrence precludes the occurrence of the primary event of interest). In observational cohort studies, these models do not allow to interpret data in terms of direct causal inference but are designed to account for competing risks [31]. Among cases, participants contributed person time until the date of T2D occurrence. Among non-cases, participants were censored at the date of last completed questionnaire, the date of diagnosis of an incident type 1 diabetes (88 cases, crude incidence rate 12.3 for 100,000 person-years,

considered as "non-cases"), the date of death, or October 1, 2021, whichever occurred first. The proportional hazard assumption was verified by using rescaled Schoenfeld-type residuals.

An age (timescale) and sex-adjusted model was first performed. Then, the main model was adjusted for age (as timescale), sex, educational level (primary, secondary, undergraduate, postgraduate), smoking status (current daily, current occasional, former, and never smokers), number of pack-years (continuous), BMI (kg/m$^2$, continuous), physical activity (high, moderate, low, calculated according to IPAQ recommendations [18]), energy intake without alcohol (kcal/d, continuous), alcohol (including restricted cubic splines to account for nonlinearity), natural sugars, added sugars, saturated fatty acids, and fiber intakes (g/d, continuous), vitamin C (mg/d, continuous), beta-carotene (μg/d, continuous), sodium (mg/d, continuous) and heme iron intakes (mg/d, continuous), number of 24-h dietary records (continuous), family history of T2D (yes/no), use of dietary supplements (yes/no), and artificial sweetener intake (mg/d) as potential markers of health consciousness, as well as the proportion in weight of ultra-processed food in the diet (as defined by the NOVA classification) [32] as an overall dietary pattern. All models were mutually adjusted for nitrate/nitrite intakes other than the specific one studied: For example, when evaluating additives-originated nitrites, we adjusted for foods and water-originated nitrites and for total nitrates. Multiple Imputation by Chained Equations method by fully conditional specifications (20 imputed datasets) was performed to handle missing data for the following covariates: physical activity level (13.9%), level of education (6.2%), BMI (0.8%), and smoking status (0.2% of missing data). Dose-response relationships were investigated graphically with restricted cubic splines. Associations between nitrites/nitrates from different food sources (fruits and vegetables, unprocessed and processed meat products) were also investigated, with mutual adjustment.

It has been suggested that antioxidant intake might be an effect-modifier of the associations between nitrites and T2D risk by affecting endogenous conversion of nitrites to nitrosamines [13]. Thus, interactions between nitrite/nitrate intakes and intakes of antioxidant vitamins (A, C, and E, coded as binary categorical variables: below/above the sex-specific median) on the other hand were tested as secondary analyses, by introducing the product of the 2 variables into the Cox models. Interactions with sex were also tested. Several models were also tested as sensitivity analyses: (1) exclusion of the first 2 years of follow-up for all participants to challenge a potential reverse causality bias; (2) adjustment for a healthy dietary pattern; (3) adjustment for sugar-sweetened beverage (SSB) consumption (g/d); (4) adjustment for prevalent hypertension, cardiovascular diseases, and hypertriglyceridemia; (5) restricting analyses to participants older than 30 years; and (6) to those having at least three 24-h dietary records. In order to estimate the relative strength of the associations, a mutually adjusted model was built including foods and water-originated nitrites, additives-originated nitrites, foods and water-originated nitrates, and additives-originated nitrates (model 7).

Since a recent study suggested positive associations between mouthwash use and T2D through an antibacterial oral effect that might interact with the nitrate-nitrite-NO pathway [33], we tested on a subsample of participants the associations between nitrates and T2D risk after adjustment and stratification for mouthwash use. We also explored using logistic regressions the cross-sectional associations with metabolic syndrome (defined as at least 3 out of 5 conditions [34]: abdominal obesity, elevated blood pressure, hypertriglyceridemia, low HDL-cholesterol, hyperglycaemia).

All methods have been described in line with the Strengthening the Reporting of Observational Studies in nutritional Epidemiology guidelines (S1 STROBE Checklist). Multi-adjusted Cox models for several confounders were prespecified. The main analyses added following the review process were as follows: further adjustments (for dietary supplement use, artificial sweeteners intake, vitamin C and beta-carotene intakes, added sugar intake, restricted cubic

splines for alcohol consumption, and dietary pattern using ultra-processed food consumption in the main model, prevalent hypertension, cardiovascular diseases and hypertriglyceridemia, and consumption of SSBs in sensitivity analyses), analyses by food sources (fruits and vegetables, red and processed meats), the mutually adjusted model, adjustment and stratification for mouthwash use in sensitivity analyses, and cross-sectional analyses with metabolic syndrome in exploratory analyses. All tests were 2 sided, and we considered $P < 0.05$ to be statistically significant. SAS version 9.4 (SAS Institute) was used for the analyses.

### Patient involvement statement

The research question developed in this article corresponds to a strong concern of the participants involved in the NutriNet-Santé cohort and of the public in general. Participants to the study are thanked in the Acknowledgements section. The results of the present study will be disseminated to the NutriNet-Santé participants through the cohort website, public seminars, and a press release.

## Results

A total of 104,168 participants (among which 72,474 [79.1%] women) were included in the present study. The mean baseline age of participants was 42.7 (SD 14.5) years (age distribution in Figure B in S1 Appendix). Table 1 shows the main baseline characteristics of participants according to their exposure to total nitrites. Compared with the first tertile (crude comparisons), participants among the highest tertile of total nitrites were more likely to be older, to have a higher BMI on average, to have family history of T2D, a higher educational degree, a higher physical activity level, and were less likely to have never smoked. Furthermore, they had higher intakes of energy, water, saturated fatty acids, sugar, alcohol, sodium, beta-carotene and vitamin C, higher consumption levels of fruit, vegetables and red and processed meats, as well as lower contribution of ultra-processed food in their diet. Foods-originated nitrites contributed to 95.3% of total nitrite intakes, followed by food additives-originated nitrites (4.7%) and water-originated nitrites (<0.01%). Foods-originated nitrates were also the main contributors to total nitrate exposure (93.0%), followed by water-originated nitrates (6.9%) and food additives-originated nitrates (0.1%). The main food groups contributing to foods-originated nitrites and nitrates were vegetables and vegetable-based meals (41% and 60%, respectively) followed by processed meat for nitrites (19%) and seasonings for nitrates (23%) (Fig 1). The main food group vector for additives-originated nitrites/nitrates was processed meat consumed as such (60% and 92%, respectively) followed by miscellaneous preparations containing processed meat (Fig 2). Some food groups contribute to both foods-originated and additives-originated nitrites and nitrates: For example, meat products are sources of both (1) foods-originated nitrates/nitrites (even before adding nitrates/nitrites additives), since animals consume themselves sources of nitrates/nitrites such as contaminated water or plant-based foods; and (2) additives-originated nitrites. Approximately, three-quarters of the study sample (73.9%) were exposed to sodium nitrite as food additive (e250) and a third (31.6%) to potassium nitrate as food additive (e252).

Participants were followed for a median of 7.3 years (interquartile range: [3.2; 10.1] years), corresponding to 710,122 person-years, during which 969 cases of T2D occurred (mean [SD] age at T2D occurrence = 59.3 [11.2] years). In age and sex-adjusted models (Table A in S1 Appendix), associations with T2D risk were observed with total nitrites ($P_{trend} < 0.001$), foods- and water-originated nitrites ($P_{trend} < 0.001$), as well as additives-originated nitrites ($P_{trend} < 0.001$) and nitrates ($P_{trend} = 0.02$) (specifically e250 and e252).

**Table 1. Baseline characteristics of the study population, NutriNet-Santé cohort, France, 2009–2021 (n = 104,168).**

| Characteristics | All participants | Categories of exposure to total nitrites* | | |
|---|---|---|---|---|
| | | Tertile 1 (n = 34,558) | Tertile 2 (n = 34,834) | Tertile 3 (n = 34,776) |
| Age at baseline, mean (SD) | 42.7 (14.5) | 40.2 (15.6) | 44.0 (14.5) | 43.7 (14.2) |
| Women (%) | 82,474 (79.1%) | 27,362 (79.2%) | 27,570 (79.1%) | 27,542 (79.2%) |
| Mean (SD) BMI (kg/m$^2$) ** | 23.7 (4.3) | 23.1 (4.1) | 23.7 (4.3) | 24.2 (4.7) |
| Family history of T2D, yes (%) $ | 14,369 (13.8%) | 4,434 (12.8%) | 4,873 (14.0%) | 5,062 (14.6%) |
| IPAQ physical activity level (%) ∞ | | | | |
| High | 29,293 (32.6%) | 9,052 (30.6%) | 9,767 (32.5%) | 10,474 (34.7%) |
| Moderate | 38,649 (43.0%) | 12,918 (43.6%) | 12,983 (43.2%) | 12,748 (42.2%) |
| Low | 21,896 (24.4%) | 7,642 (25.8%) | 7,304 (24.3%) | 6,950 (23.0%) |
| Education level (%) £ | | | | |
| < High school degree | 17,035 (16.5%) | 5,471 (16.0%) | 5,912 (17.1%) | 5,652 (16.4%) |
| <2 years after high school | 16,287 (15.8%) | 6,103 (17.8%) | 5,254 (15.2%) | 4,930 (14.3%) |
| ≥2 years after high school | 69,952 (67.7%) | 22,716 (66.2%) | 23,348 (67.6%) | 23,888 (69.3%) |
| Smoking status (%) ‡ | | | | |
| Current daily | 10,420 (10.0%) | 3,891 (11.3%) | 2,916 (8.4%) | 3,613 (10.4%) |
| Current occasional | 4,310 (4.1%) | 1,448 (4.2%) | 1,215 (3.5%) | 1,647 (4.7%) |
| Former | 42,223 (40.6%) | 12,550 (36.3%) | 14,374 (41.3%) | 15,299 (44.0%) |
| Never | 47,144 (45.3%) | 16,644 (48.2%) | 16,308 (46.8%) | 14,192 (40.8%) |
| Dietary supplement use, yes (%) | 53,857 (51.7%) | 18,013 (52.1%) | 17,934 (51.5%) | 17,910 (51.5%) |
| Mean (SD) energy intake without alcohol (kcal/d) | 1,846.2 (451.6) | 1,703.2 (403.5) | 1,859.7 (422.7) | 1,974 (483.6) |
| Mean (SD) water intake (g/d) | 2,111.7 (645.0) | 1,847.8 (573.3) | 2,120.8 (580.9) | 2,364.8 (669.0) |
| Mean (SD) alcohol intake (g/d) | 7.8 (11.8) | 5.6 (9.8) | 6.9 (10.3) | 11.0 (14.1) |
| Mean (SD) natural sugar intake (g/d) | 54.1 (21.9) | 47.7 (19.3) | 55.1 (20.2) | 59.4 (24.2) |
| Mean (SD) added sugar intake (g/d) | 38.7 (23.7) | 39.6 (24.0) | 38.7 (22.9) | 37.7 (23.9) |
| Mean (SD) fiber intake (g/d) | 19.5 (7.2) | 17.3 (6.7) | 19.7 (6.5) | 21.4 (7.7) |
| Mean (SD) saturated fatty acids (g/d) | 33.2 (12.1) | 30.5 (10.9) | 33.5 (11.5) | 35.6 (13.1) |
| Mean (SD) percentage of ultra-processed food weight in the diet | 17.3 (9.8) | 18.8 (11.0) | 16.7 (9.2) | 16.4 (8.8) |
| Mean (SD) sodium intake (mg/d) | 2,712.3 (880.9) | 2,383.4 (751.6) | 2,726.0 (803.8) | 3,025.5 (953.9) |
| Mean (SD) vitamin C intake (mg/d) | 115.8 (72.4) | 98.9 (73.5) | 117.5 (62.7) | 131.1 (76.7) |
| Mean (SD) beta-carotene intake (μg/d) | 3,446.0 (2,518.6) | 2,819.2 (2,288.2) | 3,502.2 (2,303.4) | 4,012.4 (2,787.4) |
| Mean (SD) artificial sweetener intake (mg/d) | 15.6 (48.2) | 13.9 (44.4) | 14.6 (44.2) | 18.3 (55.1) |
| Mean (SD) consumption of fruit (g/d) | 189.9 (148.0) | 144.7 (117.4) | 200.2 (137.4) | 224.4 (172.3) |
| Mean (SD) consumption of vegetables (g/d) | 217.2 (116.8) | 165.0 (90.5) | 220.9 (97.7) | 265.6 (134.5) |
| Mean (SD) consumption of red meat (g/d) | 41.7 (38.8) | 30.7 (31.7) | 43.8 (36.9) | 50.5 (44.2) |
| Mean (SD) consumption of processed meat (g/d) | 19.4 (23.8) | 15.3 (21.0) | 19.2 (21.9) | 23.8 (27.2) |
| Mean (SD) heme iron intake (mg/d) | 1.2 (1.2) | 0.8 (0.8) | 1.2 (1.1) | 1.5 (1.4) |
| Mean (SD) total nitrite intake (mg/d) | 5.7 (3.4) | 3.3 (0.9) | 5.1 (0.7) | 8.6 (4.3) |
| Mean (SD) foods-originated nitrites (mg/d) | 5.3 (3.2) | 3.1 (0.8) | 4.9 (0.8) | 8.0 (4.1) |
| Mean (SD) water-originated nitrites (mg/d) | $4 \times 10^{-4}$ ($5 \times 10^{-3}$) | $3 \times 10^{-4}$ ($4 \times 10^{-3}$) | $4 \times 10^{-4}$ ($4 \times 10^{-3}$) | $5 \times 10^{-4}$ ($6 \times 10^{-3}$) |
| Mean (SD) additives-originated nitrites (mg/d) | 0.31 (1.01) | 0.14 (0.21) | 0.22 (0.29) | 0.56 (1.68) |
| Mean (SD) sodium nitrite (e250) intake (mg/d) | 0.28 (0.86) | 0.14 (0.21) | 0.22 (0.29) | 0.47 (1.43) |
| Mean (SD) total nitrate intake (mg/d) | 213.2 (110.8) | 166.9 (85.9) | 217.1 (94.8) | 255.3 (128.4) |
| Mean (SD) foods-originated nitrates (mg/d) | 198.4 (108.7) | 153.5 (83.9) | 202.2 (93.1) | 239.1 (126.6) |
| Mean (SD) water-originated nitrates (mg/d) | 14.6 (12.3) | 13.3 (11.5) | 14.7 (12.0) | 15.9 (13.1) |
| Mean (SD) additives-originated nitrates (mg/d) | 0.18 (0.51) | 0.12 (0.37) | 0.18 (0.48) | 0.25 (0.62) |

(*Continued*)

**Table 1.** (Continued)

| Characteristics | All participants | Categories of exposure to total nitrites* | | |
|---|---|---|---|---|
| | | Tertile 1 ($n$ = 34,558) | Tertile 2 ($n$ = 34,834) | Tertile 3 ($n$ = 34,776) |
| Mean (SD) potassium nitrate (e252) intake (mg/d) | 0.18 (0.41) | 0.11 (0.30) | 0.17 (0.38) | 0.24 (0.51) |

Values are n (%) unless stated otherwise.

*P*-values were obtained using ANOVA models for continuous variables, and Chi-squared tests for categorical ones, and were all significant (<0.001), except for dietary supplement use ($p$ = 0.1).

IPAQ: International Physical Activity Questionnaire; 1 kcal = 4.18 kJ = 0.00418 MJ.

* Categories of exposure were defined as sex-specific tertiles. Tertile cutoffs for total nitrites were: 4.03 mg/d and 5.55 mg/d in women and 5.18 mg/d and 7.44 mg/d in men.

** Available for 102,580 participants.

§ Among first degree relatives.

£ Available for 103,274 participants.

∞ Available for 89,838 participants.

‡ Available for 104,097 participants.

The fully adjusted associations between intakes of nitrites or nitrates (total, foods and water-originated, additives-originated) with T2D risk are presented in Fig 3. The proportional hazard assumptions were met (Figure C in S1 Appendix). In the fully adjusted model (Fig 3), total nitrites and foods and water-originated nitrites were both positively associated with higher T2D risk (HR$_{\text{tertile 3 vs.1}}$ = 1.27 (95% CI 1.04 to 1.54), P$_{\text{trend}}$ = 0.009 and 1.26 (95% CI 1.03 to 1.54), P$_{\text{trend}}$ = 0.02, respectively). Participants with higher exposure to nitrites from

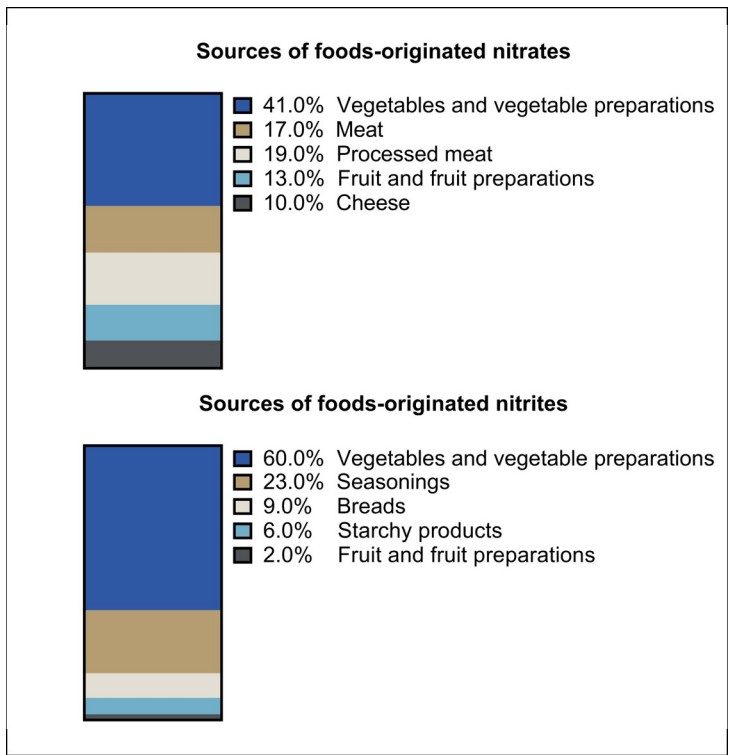

**Fig 1. Food sources of foods-originated nitrites and nitrates, NutriNet-Santé cohort, France 2009–2021 ($n$ = 104,168).**

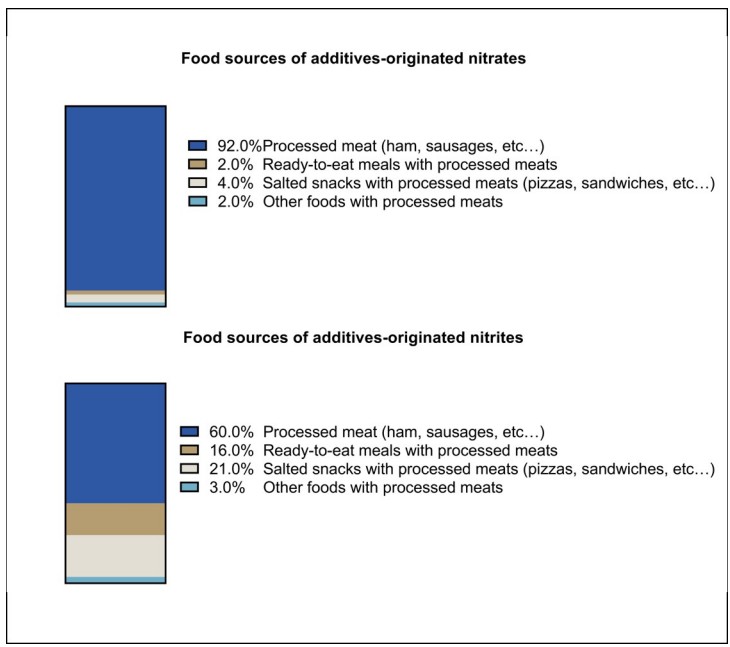

**Fig 2. Food sources of nitrites and nitrates as food additives, NutriNet-Santé cohort, France 2009–2021 (*n* = 104,168).**

food additives (i.e., above the sex-specific median), and specifically those having higher exposure to sodium nitrite (e250) had a higher T2D risk compared with those who were not exposed to food additive nitrites (HR$_{\text{higher consumers vs. non-consumers}}$ = 1.53 (95% CI 1.24 to 1.88), P$_{\text{trend}}$ < 0.001 and 1.54 (95% CI 1.26 to 1.90), P$_{\text{trend}}$ < 0.001, respectively). Dose-responses relationships (Fig 4) showed no evidence of nonlinearity with total and foods and water-originated nitrites (*p* = 0.08 and 0.1, respectively). As regards nitrites from food additives, the curve showed a nonlinear shape with a plateau in the highly exposed participants (p for nonlinearity <0.001). There was no association between total, foods and water-originated, or additives-originated nitrates and T2D risk (all P$_{\text{trend}}$ = 0.7).

Analyses from food sources showed statistically signification associations for nitrites from red and processed meats (HR$_{\text{tertile 3 vs.1}}$ = 1.30 (95% CI 1.30 to 1.58), P-trend = 0.01) and nitrates from red and processed meats (HR$_{\text{tertile 3 vs.1}}$ = 1.34 (95% CI 1.10 to 1.63), P-trend = 0.03), but not for nitrites from fruits and vegetables (HR$_{\text{tertile 3 vs.1}}$ = 1.10 (95% CI 0.88 to 1.37), P-trend = 0.4) or nitrates from fruits and vegetables (HR$_{\text{tertile 3 vs.1}}$ = 0.92 (95% CI 0.74 to 1.14), P-trend = 0.5) (Table B in S1 Appendix).

Evidence from interactions with antioxidant intakes was inconclusive. In some cases, associations tended to be more pronounced in participants with antioxidants intake above the median; however, interaction terms were not significant (Table C in S1 Appendix).

Despite nonsignificant interactions with sex (*p* > 0.1), sex-specific models were performed: in women, associations were found between total nitrites (HR$_{\text{tertile 3 vs. 1}}$ = 1.47 (95% CI 1.14 to 1.90), P$_{\text{-trend}}$ = 0.002), nitrites from additives (HR$_{\text{higher consumers vs. non-consumers}}$ = 1.36 (95% CI 1.06 to 1.75), P$_{\text{-trend}}$ = 0.006) and foods and water-originated nitrites ((HR$_{\text{tertile 3 vs. 1}}$ = 1.50 (95% CI 1.15 to 1.94), P$_{\text{-trend}}$ = 0.002). In men, associations were only significant for nitrites from food additives (HR$_{\text{higher consumers vs. non-consumers}}$ = 1.65 (95% CI 1.15 to 2.36), P$_{\text{-trend}}$ = 0.009) (Table C in S1 Appendix).

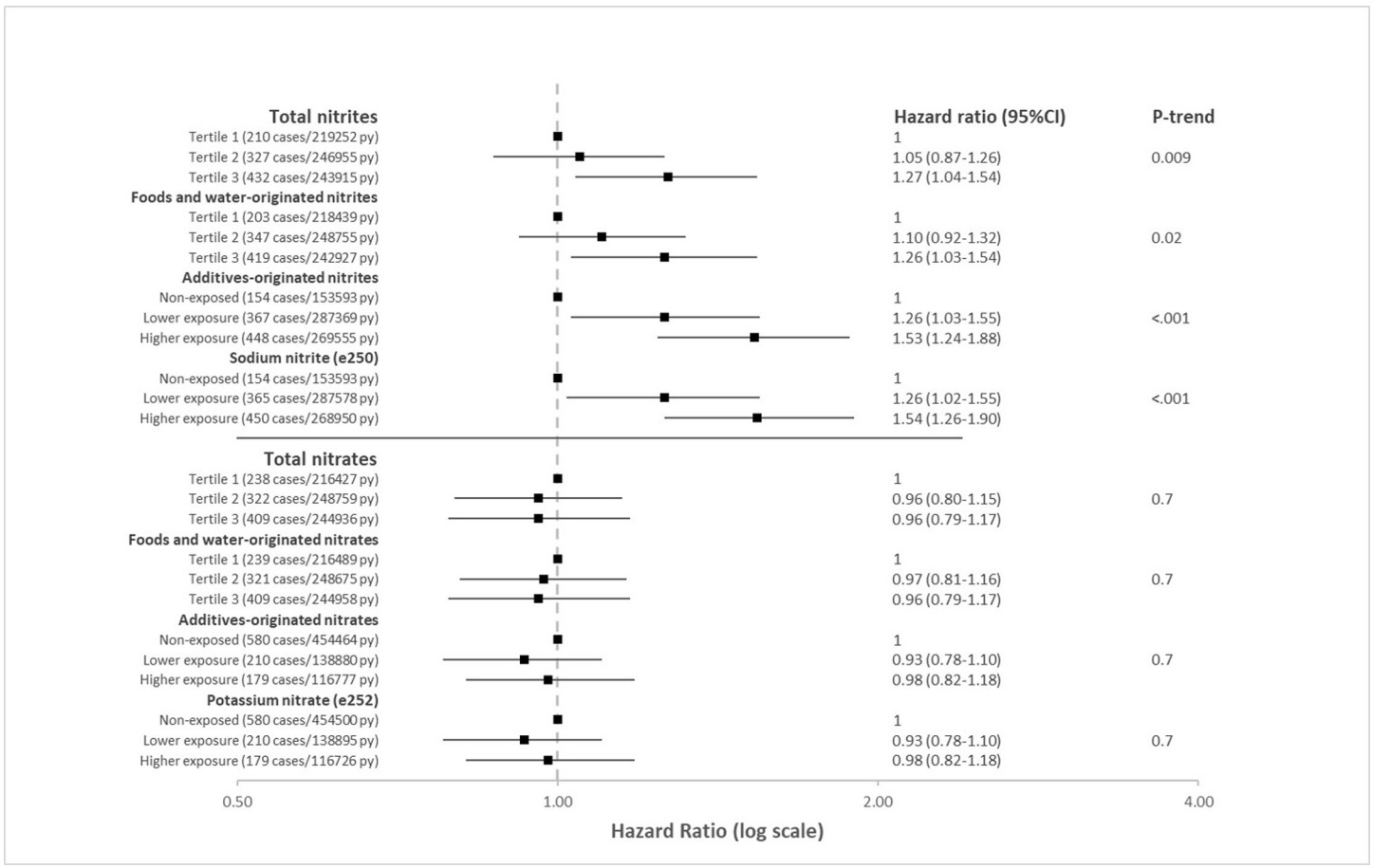

**Fig 3. Forest-plots of the associations between dietary exposure to nitrites and nitrates with T2D risk, NutriNet-Santé cohort, France, 2009–2021 (*n* = 104,168).** HR: hazard ratio; 95% CI: 95% confidence interval; PY: person-years. Multivariable Cox proportional hazard model were adjusted for: age (time scale), sex, energy intake without alcohol (kcal/d, continuous), alcohol (including restricted cubic splines to account for nonlinearity), sodium, natural sugars, added sugars, saturated fatty acids, and fiber intakes (g/d, continuous), heme iron intakes (mg/d, continuous), vitamin C intakes (mg/d, continuous), beta-carotene intakes (mg/d, continuous), BMI (kg/m², continuous), physical activity (high, moderate, low, calculated according to IPAQ recommendations), smoking status (never, former, current daily, current occasional smokers), number of pack-years, number of 24-h dietary records (continuous), family history of diabetes (yes/no), educational level (<high school degree, <2 years after high school, ≥2 years after high school), dietary supplement use (yes/no), artificial sweetener intake (mg/d), proportion of UPF in the diet. All models were mutually adjusted for nitrate/nitrite intakes other than the specific one studied. For example, when evaluating additives-originated nitrites, we adjusted for foods and water-originated nitrites and for total nitrites. For exposure to total nitrites and nitrates and foods and water-originated nitrites and nitrates, sex-specific tertiles of exposure were defined. Cut-offs were: 4.03 mg/d and 5.55 mg/d in women and 5.18 mg/d and 7.44 mg/d in men for total nitrites, 150.09 mg/d and 233.89 mg/d in women and 162.11 mg/d and 251.59 mg/d in men for total nitrates, 3.83 mg/d and 5.29 mg/d in women and 4.92 mg/d and 7.07 mg/d in men for foods and water-originated nitrites, 149.91 mg/d and 233.75 mg/d in women and 161.94 mg/d and 251.32 mg/d in men for foods and water-originated nitrates. For additives-originated nitrites and nitrates, 3 categories of exposure were defined: non-exposed, lower exposure, and higher exposure (separated by sex-specific median among exposed participants). Cut-offs were: 0.19 mg/d in women and 0.25 mg/d in men for additives-originated nitrites, 0.36 mg/d in women and 0.46 mg/d in men for additives originated nitrates, 0.19 mg/d in women and 0.25 mg/d in men for sodium nitrite (e250), 0.36 mg/d in women and 0.46 mg/d in men for potassium nitrate (e252). Cause-specific associations with all-cause mortality as a competing risk are presented in Table G in S1 Appendix.

Overall, results remained similar in all sensitivity analyses or were slightly attenuated (Table D in S1 Appendix). In the mutually adjusted model (model 7), associations were detected for foods and water-originated nitrites (HR$_{tertile\ 3\ vs.1}$ = 1.22 (95% CI 1.00 to 1.49), P$_{-trend}$ = 0.05), and additives-originated nitrites (HR$_{higher\ consumers\ vs.\ non-consumers}$ = 1.48 (95% CI 1.21 to 1.82, P$_{-trend}$ = <0.001), but not for foods and water-originated or additives-originated nitrates (P$_{-trend}$ >0.5) (Table D in S1 Appendix).

Furthermore, no associations between nitrates and T2D risk were observed independently of mouthwash use (Table E in S1 Appendix).

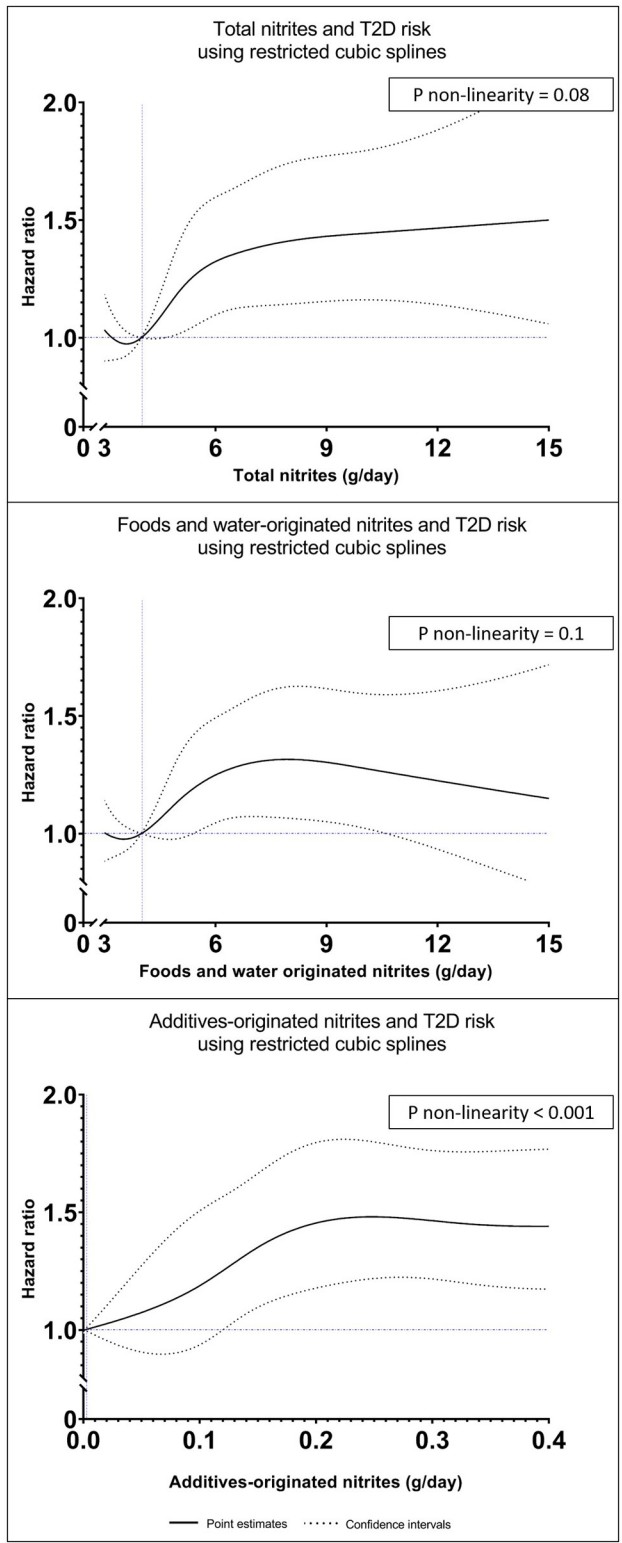

**Fig 4. Dose-response relationships between dietary nitrites (total, foods and water-originated, and additives-originated) and T2D risk, NutriNet-Santé cohort, 2009–2021 (*n* = 104,168).**

In the exploratory cross-sectional analysis with metabolic syndrome, statistically significant associations were found for total nitrites (OR $_{\text{tertile 3 vs. 1}}$ = 1.35 (95% CI 1.16 to 1.58), P$_{\text{-trend}}$ <0.001), and foods and water-originated nitrites (OR $_{\text{tertile 3 vs. 1}}$ = 1.45 (95% CI 1.24 to 1.71), P$_{\text{-trend}}$ <0.001), but not for additives-originated nitrites (OR $_{\text{tertile 3 vs. 1}}$ = 1.06 (95% CI 0.90 to 1.25), P$_{\text{-trend}}$ = 0.5), or nitrates (P$_{\text{-trend}}$ >0.1) (Table F in S1 Appendix). However, the evidence that can be drawn out of these findings is weak, given the very restricted sample size and the absence of a prospective design.

## Discussion

This large-scale prospective cohort study highlighted an association between the exposure to nitrites, and for the first time to our knowledge, those from food additives (mostly sodium nitrite e250), with T2D risk. Total, foods and water-originated, and additives-originated nitrates were not associated with T2D risk in this study.

To our knowledge, only 1 prospective study conducted in Iran [13] explored the associations between dietary exposure to nitrites and nitrates and T2D risk, with no distinction made between foods and water origin and food additives origin. This study included 2,139 adults followed for a median of 5.8y. and suggested positive associations between dietary exposure to total and animal-based nitrites and T2D risk, only in participants with low vitamin C intakes, but not in the whole sample. The Iranian study assessed diet using food frequency questionnaires, which are recognized to provide reliable estimates of dietary habits in terms of "generic" food group consumption [35]. Dietary records are also recognized as appropriate tools to capture usual dietary habits, when administered in a repeated manner (as done in the present study: mean number of records/participant = 6, maximum = 18) [36,37]. Besides, records collecting brand-specific information for commercial products provide higher levels of accuracy to evaluate exposure to specific food additives coming from several commercial products with different brands (given the substantial variation of additive content between commercial brands). In our study, interactions with antioxidants intakes were mostly nonsignificant. In some cases, the associations we observed between nitrites with T2D risk appeared to be more pronounced in participants with higher vitamin C, A, and E intakes (these results should be interpreted with caution since interaction tests were not significant). Several hypotheses could explain these results in our population: first, fruit and vegetables are an important source of both nitrites and vitamin C. Therefore, individuals in the top median of vitamin C intakes are also those with higher values of nitrites intake (mean food and water-originated nitrite intake = 5.8 mg/d versus 4.9 mg/d in participants with vitamin C intake below the median), thus, we were more inclined to detect associations between nitrite intake and T2D risk in this subgroup. Second, participants with higher antioxidant intakes were older (43.6 years in the top vitamin C category versus 41.7 in the bottom category) and had higher energy intakes (1,934 Kcal/day in the top vitamin C category versus 1,757 in the bottom category), which has led to a higher number of cases and a stronger statistical power in participants with higher antioxidant intakes. In the same study by Bahadoran and colleagues [13], no association was observed between dietary nitrates and T2D risk, consistently with our results. However, median intakes of nitrites and nitrates in the Iranian study were higher than those in our study (410 mg/d versus 194 mg/d in our study for nitrates and 8.8 mg/d versus 5.0 mg/d in our study for nitrites). Green leafy vegetables are the major source of dietary nitrates. Even though their role in reducing T2D risk has been suggested [38,39], the most recent meta-analyses [40–42] have reported low to very low quality of evidence for the association of cruciferous and green leafy vegetables with T2D risk. This is in line with our findings regarding the lack of association between foods-originated nitrates with T2D risk, or fruits and vegetables-

originated nitrites and nitrates with T2D risk. In any case, even though the evidence from our study for these specific analyses might be relatively weak given the (1) moderate-to-strong correlations between nitrites and nitrates from fruits and vegetables with their other components (e.g., vitamin C [Pearson correlation coefficients = 0.41 for nitrites and 0.31 for nitrates], beta-carotene [Pearson correlation coefficients = 0.50 for nitrites and 0.58 for nitrates], dietary fibers [Pearson correlation coefficients = 0.63 for nitrites and 0.57 for nitrates]); (2) the possible lack of statistical power; and (3) potential residual confounding or classification biases, vegetables remain important sources of fibers and antioxidants, and are beneficial components of a healthy diet playing an important role in the prevention of chronic diseases such as cardiovascular diseases and some cancers. Previously published meta-analyses [40,43] reported high evidence levels for the role of processed meat, an important contributor to intakes of nitrites as food additives, as a risk factor for T2D risk. Although it is not possible to fully disentangle the effects of each processed meat constituent, the fact that our models were adjusted for others components that could play a role in T2D aetiology (e.g., pro-oxidant heme iron, saturated fatty acids, sodium) suggests that nitrites contained in processed meat might causally contribute to processed meat–T2D associations, which is in line with dietary guidelines recommending to reduce red and processed meats consumption.

Mechanistically, our associations support the suggested role of dietary nitrites observed in experimental studies [11,12], through *N*-nitroso compounds, in the development of insulin resistance via a disruption of insulin and IGF signaling pathways, and a dysfunction of pancreatic β cells. Moreover, an experimental study on rodent models [44] suggested that long-term high intakes of nitrates or acute high intakes of nitrites down-regulate the eNOS activity (the latter playing a major role in vascular and tissue protection), suggesting a negative cross-talk between the NOS pathway with endogenous nitrates and the nitrate-nitrite-NO pathway with dietary nitrates, which might have deleterious health outcomes. Furthermore, a recent study suggested positive associations between regular mouthwash use and T2D through an antibacterial oral effect that might stop the reduction of nitrates to nitrites in the oral cavity [33]. Our results showed no association with nitrates, independently of regular mouthwash use. However, data on mouthwash use was available only in a small subgroup of participants; therefore, we might have lacked statistical power to accurately investigate this assumption. In addition, only 15% of the sample reported using mouthwash at least once a week. As regards the analysis with metabolic syndrome showing a significant positive association with total and foods and water-originated but not with additives-originated nitrites, this model followed a cross-sectional design since biological measures were only available once; thus, the temporal sequence between the exposure and the outcome was not respected. These findings might be due to a reverse causation bias, as participants with a metabolic syndrome could have modified their diets by increasing fruit and vegetable consumption and reducing processed meats.

This prospective study includes several strengths: to our knowledge, it is the first to have explored the associations between nitrites/nitrates, separately from foods and water, and food additives, with T2D risk, using a large panel of confounders and a detailed comprehensive dietary assessment. We used repeated 24-h records to collect information on a wide range of foods containing nitrates and nitrites while specifying their origin (foods and water or food additives). We collected specific information about the consumed commercial brands/names, which is crucial given the heterogeneity of food additives between the available brands. We also used data from 3 different databases coupled to thousands of assays, as well as regional and temporal information about water contamination.

However, some limitations should be acknowledged. First, causation could not be established from this single observational study and residual confounding could not be entirely ruled out. Nevertheless, we adjusted for a wide range of sociodemographic, anthropometric,

lifestyle, dietary, and health factors in order to limit this potential bias. Second, even though dietary records were validated against blood and urinary biomarkers for energy and key nutrients, exposure to nitrates and nitrates has not been validated against blood or urine assays due to lack of biomarkers that would be specific enough to reflect exogenous dietary exposure and not endogenous metabolism. For example, plasma nitrate would not only be a marker of dietary sources, but would also reflect nitrate derived from various individual (patho-) physiological pathways, as well as the individual capacity to eliminate nitrate [45–47]; similarly for urinary nitrites, in addition to urinary tract infections being an important driver of urine nitrites levels [48] and could provide biased estimates of dietary exposure. Therefore, and despite the use of multiple databases and detailed dietary assessment, the validity of the exposure assessment cannot be guaranteed, the sensitivity and specificity are unclear. Although likely non-differential due to the prospective design, these potential measurement errors may have biased the associations towards an unclear direction. Future validations studies could be considered; for example, using a duplicate-meal design for multiple days to estimate habitual nitrate/nitrite intakes or biomarkers from tissues.

Moreover, as in every population-based observational cohort, generalizability of the results to the whole population should be discussed. Compared with the general French population, NutriNet-Santé participants were younger, more often women, with higher educational and socio-professional levels [49]. NutriNet-Santé shows a good representativeness of adults aged below 75 years old, as this population has mostly home access to the internet [50], and under-represents participants above 75 years old, who have less access to regular home internet connection. Therefore, caution is needed when extrapolating results from this cohort to participants aged above 75 years old. NutriNet-Santé's participants were less likely to smoke [51] and to be overweight or obese (28.2% of men and 29.4% of women in NutriNet-Santé versus 54% and 44% in French population [51]). These aforementioned factors explain the lower T2D incidence in our cohort (179 cases per 100,000 person-years after age and sex standardization) in comparison with the general French population (289 cases per 100,000 person-years [52]). Participants had also healthier dietary habits [53], as they had higher consumption levels of fruit and vegetables and lower levels of processed meats consumption compared to the general population, which led to higher levels of exposure to total nitrites and nitrates and lower levels of exposure to food additives in NutriNet-Santé, compared with nationally representative surveys: indeed, exposure levels in the French INCA2 nationally representative survey were 0.04 mg/kg of BW for total nitrites [25] and 1.6 mg/kg BW for total nitrates [26], versus 0.09 and 3.34 mg/kg BW in NutriNet-Santé, respectively. Similarly, average intakes of nitrates in our study were higher than those reported in a systematic review by Babateen and colleagues [54]: average intake of total nitrates in healthy European populations = 107.21 mg/d versus 213.2 mg/d in our study. In contrast, median intakes of nitrites and nitrates in an Iranian study were higher than those in our study (410 mg/d versus 194 mg/d in our study for nitrates and 8.8 mg/d versus 5.0 mg/d in our study for nitrites) [13]. As regards food additives specifically, exposure levels were 0.01 to 0.04 mg/kg BW for nitrites from additives and 0.05 to 0.10 mg/kg BW for nitrates from additives in the multi-country EFSA report, versus 0.004 mg/kg BW and 0.003 mg/kg BW in NutriNet-Santé, respectively. This might have led to an underestimation of true associations with nitrites and nitrates from food additives (due to the reduced contrast between compared groups). In contrast, regarding foods and water-originated nitrites and nitrates, the distribution of exposure in our population allowed us to have enough high consumers to properly explore this relatively extreme part of the curve, which exists (to a lesser extent) in the general population. As classically observed in nutritional epidemiology studies, a significant subsample of the cohort (17%) was flagged as energy under-reporters and were excluded from the final sample. In the nationally representative INCA 3 study conducted in

2016 by the French Food Safety Agency [55], 18% of adult participants were identified as under-reporters using the Black method. Under-reporters in our study were older and were more inclined to be male and current smokers, to have a higher BMI and alcohol intake, and a lower educational level and monthly income. Although their exclusion may limit the generalizability of the findings, it was necessary in order to avoid important exposure classification bias.

Furthermore, with the continuous reformulation of industrial products, assessing the exposure from food additives might be complicated. However, we proceeded with a year-specific dynamic matching, accounting for different compositions of a same product/brand based on the year of consumption. In addition, although under-detection of some diagnosed T2D cases is still possible at baseline and during follow-up, it is likely to have been limited thanks to the multisource strategy for case ascertainment (combining self-report of disease, self-report of medication, link to national health insurance database, and fasting blood glucose on a subsample). The fact that exhaustiveness of T2D case detection cannot be guaranteed (especially those that have never been diagnosed or treated) would mainly have led to reduced statistical power and non-differential classification errors and thus likely to an underestimation of the observed associations. In contrast, it is not likely to explain the statistically significant associations observed. Lastly, classification bias could be possible while estimating water-originated exposure to nitrites/nitrates based on regional and temporal information about tap water contamination, since some individuals may have spent time outside their region of residence. However, French national statistics show that the average French person spends less than a month (27.9 days) outside their residence [56]. Moreover, water-originated nitrites and nitrates represent a small minority of foods and water-originated nitrites and nitrates intakes (<0.01% and 6.9%, respectively). This potential bias was therefore not likely to have a significant effect on our estimates.

In this large prospective cohort, a higher exposure to foods and water-originated and additives-originated nitrites was associated with higher T2D risk. Thus, these results did not support any potential benefits for dietary nitrites or nitrates in T2D prevention. These findings need confirmation by other prospective studies and experimental research, given that the study sample overrepresented women, younger participants, and those having healthier lifestyle habits and higher socio-professional levels, and given the possible exposure measurement errors. Yet, these results provide a new piece of evidence in the context of current discussions regarding the need for a reduction of nitrite additives' use in processed meats by the food industry and could support the need for better regulation of soil contamination by fertilizers, as highlighted by the latest report of the French Agency for Food, Environmental and Occupational Health and Safety [57]. In the meantime, several public health authorities worldwide already recommend citizens to limit their consumption of foods containing controversial additives, among which sodium nitrite, in the name of the precautionary principle [58].

## Supporting information

**S1 STROBE Checklist. STROBE Checklist.**
(DOCX)

**S1 Appendix. Supplementary material.** Method A. Methodology for identification of underenergy reporting and validation studies for the 24-h web-based dietary records. Method B. Procedures for the computation of food additive data. Method C. Incident T2D ascertainment in NutriNet-Santé and biological data assessment. Figure A. Flowchart for sample selection, NutriNet-Santé, 2009–2021. Figure B. Age distribution of study participants, NutriNet-Santé cohort, 2009–2021 (*N* = 104,168). Table A. Age and sex-adjusted models for associations between nitrite and nitrate exposures and T2D risk, NutriNet-Santé cohort, France, 2009–

2021 (*n* = 104,168). Figure C. Proportional hazard assumption testing using rescaled Schoenfeld residuals. Table B. Associations between nitrite and nitrate exposures from fruit and vegetables, and red and processed meats, and T2D risk, NutriNet-Santé cohort, France, 2009–2021 (*n* = 104,168). Table C. Sex and antioxidant-stratified associations between exposure to nitrites/nitrates and T2D risk, NutriNet-Santé cohort, 2009–2021 (*n* = 104,168). Table D. Associations between nitrite and nitrate exposures and T2D risk-sensitivity analyses, NutriNet-Santé cohort, France, 2009–2021 (*n* = 104,168). Table E. Associations between dietary exposure to nitrates with T2D risk, adjusted stratified for mouthwash use, France, 2009–2021 (*n* = 25,328). Table F. Cross-sectional associations between dietary exposure to nitrites and nitrates with metabolic syndrome prevalence, NutriNet-Santé cohort, France, 2009–2021 (*n* = 16,450). Table G. Cause-specific associations between dietary exposure to nitrites and nitrates with mortality risk as a competing risk, NutriNet-Santé cohort, France, 2009–2021 (*n* = 104,168).
(DOCX)

## Acknowledgments

We thank Thi Hong Van Duong, Régis Gatibelza, Jagatjit Mohinder, Rizvane Mougamadou, and Aladi Timera (computer scientists); Julien Allegre, Nathalie Arnault, Laurent Bourhis, Nicolas Dechamp, Guillaume Javaux (data-manager/statisticians); Paola Yvroud (health event validator); Maria Gomes (Nutrinaute support) for their technical contribution to the Nutri-Net-Santé study. We warmly thank all the volunteers of the NutriNet-Santé cohort.

## Author Contributions

**Conceptualization:** Bernard Srour, Eloi Chazelas, Cédric Agaësse, Serge Hercberg, Mathilde Touvier.

**Data curation:** Bernard Srour, Eloi Chazelas, Nathalie Druesne-Pecollo, Fabien Szabo de Edelenyi, Cédric Agaësse, Alexandre De Sa, Rebecca Lutchia, Laury Sellem.

**Formal analysis:** Bernard Srour.

**Funding acquisition:** Mathilde Touvier.

**Investigation:** Bernard Srour, Charlotte Debras, Inge Huybrechts, Mathilde Touvier.

**Methodology:** Bernard Srour, Nathalie Druesne-Pecollo, Laury Sellem, Inge Huybrechts, Chantal Julia, Emmanuelle Kesse-Guyot, Benjamin Allès, Pilar Galan, Serge Hercberg, Fabrice Pierre, Mélanie Deschasaux-Tanguy, Mathilde Touvier.

**Project administration:** Nathalie Druesne-Pecollo, Younes Esseddik, Mathilde Touvier.

**Resources:** Mathilde Touvier.

**Supervision:** Mathilde Touvier.

**Validation:** Mathilde Touvier.

**Visualization:** Bernard Srour.

**Writing – original draft:** Bernard Srour.

**Writing – review & editing:** Bernard Srour, Eloi Chazelas, Nathalie Druesne-Pecollo, Younes Esseddik, Fabien Szabo de Edelenyi, Cédric Agaësse, Alexandre De Sa, Rebecca Lutchia, Charlotte Debras, Laury Sellem, Inge Huybrechts, Chantal Julia, Emmanuelle Kesse-Guyot,

Benjamin Allès, Pilar Galan, Serge Hercberg, Fabrice Pierre, Mélanie Deschasaux-Tanguy, Mathilde Touvier.

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
