## [Editor Report · Decision Letter 0]

30 Jun 2022

Dear Dr Srour, 

Thank you for submitting your manuscript entitled "Nitrites and nitrates dietary exposure from natural sources and food additives and type-2 diabetes risk in the NutriNet-Santé cohort" for consideration by PLOS Medicine.

Your manuscript has now been evaluated by the PLOS Medicine editorial staff and I am writing to let you know that we would like to send your submission out for external peer review.

Please re-submit your manuscript within two working days, i.e. by Jul 04 2022 11:59PM.

Kind regards,

Callam Davidson

Associate Editor

PLOS Medicine

---

## [Decision Letter · Decision Letter 1]

1 Aug 2022

Dear Dr. Srour,

Thank you very much for submitting your manuscript "Nitrites and nitrates dietary exposure from natural sources and food additives and type-2 diabetes risk in the NutriNet-Santé cohort" (PMEDICINE-D-22-02217R1) for consideration at PLOS Medicine. 

Your paper was evaluated by an associate editor and discussed among all the editors here. It was also discussed with an academic editor with relevant expertise, and sent to independent reviewers, including a statistical reviewer. The reviews are appended at the bottom of this email and any accompanying reviewer attachments can be seen via the link below:

[LINK]

In light of these reviews, I am afraid that we will not be able to accept the manuscript for publication in the journal in its current form, but we would like to consider a revised version that addresses the reviewers' and editors' comments. Obviously we cannot make any decision about publication until we have seen the revised manuscript and your response, and we plan to seek re-review by one or more of the reviewers. 

We hope to receive your revised manuscript by Aug 22 2022 11:59PM. Please email us (plosmedicine@plos.org) if you have any questions or concerns.

We look forward to receiving your revised manuscript. 

Sincerely,

Callam Davidson, 

PLOS Medicine

plosmedicine.org

Comments from the Academic Editor:

The authors conducted novel observational analyses. Besides the comments from the other reviewers, the following comments should be accounted for to identify the appropriateness of this manuscript for potential publication.

The major comments are as follows:

The authors should revise the covariate set to adjust for in their regression analyses. Some concerns are present.

1-1. First, an association of alcohol consumption with T2D is known to be non-linear. The authors should fit restricted cubic spline terms.

1-2. Second, the authors should control for sugar-sweetened beverage (SSB) consumption. It is unlikely to be a source of nitrates and nitrites, unless some products uniquely contain those as ingredients, so its adjustment should be fine. Similarly or alternatively, the authors should control for added sugars from beverages and added sugars from solid foods (e.g. cakes). Sugars naturally occurring in foods and added sugars may have differential associations with T2D, as implicated in studies on SSB. Moreover, this current study is relevant to food processing or additives, and thus separation between added sugars and natural sugars is critical in covariate adjustment. In general, authors should be equally careful in exposure and covariate treatments.

1-3. Third, when the authors evaluated naturally-occurring nitrites and nitrites from food additives, they should adjust for those mutually. It was great that the authors clarified the application of mutual adjustment between nitrates and nitrites. This care is necessary for the analysis of sources of nitrates and nitrites. The authors should consider a time-to-event regression model to include nitrites from natural sources, nitrites from additives, nitrates from natural sources, and nitrates from additives.

1-4. The authors should control for intakes of vitamin C and carotenoids from foods, at least. Those intakes or biomarkers are known to be associated with incident T2D. They are likely confounders influencing the potential effect of fruits and vegetables that provide nitrates or nitrites.

1-5. The authors should control for supplement use, artificual sweetener intakes, and others which the authors may come up with as possible indicators of health consciousness. In this study, health consciousness remains a source of unmeasured confounders, and the authors must want to minimize the effect as much as possible.

2. The authors should evaluate nitrates and nitrites from different food sources in another way additionally. As expected, those chemicals mainly come from meat/processed meat consumption, regardless of how the authors identified or defined natural sources or food additives. In general, the positive associations of meat/processed meat consumption with incident T2D have been well-recognized. If those food groups were major drivers of the observed associations of nitrites, the inference from this study would not impact general dietary recommendations.

To clarify if that is the case, the authors should evaluate, for example, nitrites from fruits and vegetables and nitrites from unprocessed and processed meat products separately, with mutual adjustment.

Of note, if nitrite intake from fruits and vegetables showed positive associations with T2D incidence, that would be a meaningful finding that scientists should get aware of for further research.

The authors may find that nitrite intakes from fruits and vegetables showed no significant associations or negative associations (after adjustment for vitamin C, carotenoids, nitrites from other sources, and nitrates from different sources). This finding would be important to clarify no evidence for harm from nitrites in fruits and vegetables.

Collinearity is likely between nitrites from fruits and vegetables, vitamin C, and carotenoids. The authors should present results for the indication, if any, and state the weaknesses of the evidence for nitrites/nitrates from fruits and vegetables.

3.

In the presentation of hazard ratios, the authors should replace the numbers of participants (denominator) with person-time values. The authors evaluated hazard ratios which have close relevance to incidence (N events/ person-time) more than risk (N events/ specific time length). Thus, the authors should show person-time values to allow interpretations of incidence measures.

4.

The authors should present the results from cubic spline analyses and food sources of nitrites and nitrates in the main manuscript, not in the supplementary materials. Those are critical information to present.

Minor comments:

Table 1. The authors can take out the column of P-value and mention that all trends were significant in the footnote.

Please revise your title according to PLOS Medicine's style. Your title must be nondeclarative and not a question. It should begin with main concept if possible. "Effect of" should be used only if causality can be inferred, i.e., for an RCT. Please place the study design ("A randomized controlled trial," "A retrospective study," "A modelling study," etc.) in the subtitle (ie, after a colon).

Abstract Methods and Findings:

* Please include the length of follow up (information found at line 260).

Lines 83-86: Please confirm the correct HR is presented in this bullet point as the abstract suggests it ought to be ‘1.60 (95% CI 1.29-54 1.98), P¬trend<0.001’ for e250 specifically. 

Citations should be placed within square brackets.

Line 273: Please round p values to 3 decimal places (state p<0.001).

Table 1 and Figure 1: Footnote ‘*’ is missing the corresponding flag in the table/figure. 

For the adjusted analyses in Figure 1, please also present the crude (unadjusted) analyses in the Supporting Information and cite them in your results. 

Figure 1 legend should refer to Appendix 9 rather than 8. 

Discussion: Please ensure any claims of primacy are tempered by the addition of ‘to our knowledge’, or similar. 

Pages 17-19: Please remove the ‘Funding’, ‘Contributorship statement and guarantor’, ‘Transparency statement’, and ‘IARC disclaimer’ subsections – this information will be captured via the submission form and presented as metadata in the event of publication. The ‘Patient involvement statement’ can be relocated to the Methods section. 

Appendix 6: Please enlarge the labels for your axes and your title font. Please also refer to p<0.001 rather than P=0.0009.

Please ensure that the study is reported according to the STROBE guideline, and include the completed STROBE checklist as Supporting Information. Please add the following statement, or similar, to the Methods: "This study is reported as per the Strengthening the Reporting of Observational Studies in Epidemiology (STROBE) guideline (S1 Checklist)."

Did your study have a prospective protocol or analysis plan? Please state this (either way) early in the Methods section.

To an extent that is reasonably practical, please avoid textual overlap with your prior publication (Chazelas et al., Int J Epidemiol, 2022; https://academic.oup.com/ije/advance-article/doi/10.1093/ije/dyac046/6550543).

Comments from the reviewers:

Reviewer #1: This is an interesting manuscript analyzing the prospective association between dietary nitrites and nitrates from natural sources and food additives and type 2 diabetes in the NutriNet-Sante cohort with a huge sample size. Overall the manuscript is clear and well written, delivering important public health relevant message about the health effect of dietary nitrites and nitrates especially from the food additives. This reviewer has several comments to help improve the manuscript:

1) Exposure assessment. The reviewer appreciates their effort to validate the web-based 24h dietary records. It seems that the specifical validation for the nitrites and nitrates is lacking? Is there any way to validate these measurements, say, against biomarkers? The results were based on "at least 2 valid 24 dietary records during their first t years of follow-up", so it might be rather random.

2) Outcome ascertainment. It seems that this was based on the self-reported data. It is not clear about the "biological data". Do they provide blood sample for the measurement of glucose or HbA1c? Would underreporting at baseline be possible?

3) Confounders. They include different covariates in the models, which is great. But this reviewer believes that there are several more important confounders for the associations. For examples, those did not consume food additives/nitrites are more likely to have healthy lifestyle/dietary quality and lower risk of diabetes. Therefore, adjustment of dietary pattern and dietary quality seems an important step in the main statistical model. In the sensitivity analysis, they additional adjusted for antioxidant intakes as a proxy for fruit and vegie intake. Why not directly adjust fruit and vegie, which is also indicator of lifestyle?

4) Discussion. They should avoid using the word "robust", as they did not have external validation to prove the robustness. They did not adjust for important lifestyle/ diet quality confounders.

Reviewer #2: In the study entitled: Nitrites and nitrates dietary exposure from natural sources and food additives 

and type-2 diabetes risk in the NutriNet-Santé cohort, the authors aimed to investigate the associations between the amount of nitrite and nitrate intakes and their specific food sources (i.e., natural versus additive food sources) and new-onset diabetes, using data of 104168 adults from the French NutriNet-Santé cohort study (2009-2021). They finally concluded that regardless of its sources, higher intake of nitrite exhibited an adverse effect on the development of type 2 diabetes (T2D). No interaction was detected between exposure to nitrate/nitrite and antioxidants (Vitamin C, E and A) in relation to developing T2DM. Use of a national database (i.e., NutriNet-Santé food composition database) for estimation of nutrients, esp. nitrate and nitrites, and assessment of details of commercial names/brand of industrial products to estimate individual additive exposure provides a strong support for estimation of nitrate and nitrites in this population. Although the study design seems well and a relatively-large sample size and use of national databases of food composition are the important strengths of the study, some issues (listed below) may decrease generalizability of the study findings and its reliability. 

Major comments:

1. A low incidence rate of T2D, i.e., lower than 1% over a median of 7.3 years of follow-up (969 cases from 104168 subjects) needs explanation. In addition, the prevalence of T2D is lower in this population even compared to French population (1.6% vs. 6%) as authors acknowledged in their previous study (BMJ 2019;365:l1451; doi: 10.1136/bmj.l1451). Therefore, it should be justified that this cohort is representative of the population. This needs for generalization of data.

2. Another concern about the study findings is that the main outcome was self-reported, however the authors claimed that self-report of T2D was tried to be confirmed by supportive data such as National health insurance (SNIIRAM) database or measurement of fasting blood glucose in a sub-sample of the participants (the number of participants with FBS was not reported). Nevertheless, SNIIRAM database only confirmed T2D diagnosis of about 80% of the cases surveyed, and among those who had biochemical measurements, only 85.3 % of those with elevated fasting blood glucose had consistently reported a diagnosis of T2DM and/or medication. Reporting bias of the outcome in this study seem resulted in a considerable underestimation of the outcome that seriously may affect validity and reliability of the study findings. This important issue, therefore, should be discussed by providing a number of relevant evidence indicated that whether such limitation can affect the study findings validity. They should also support their findings by providing national data regarding the prevalence and annual incidence if diabetes in their country. I wonder if undiagnosed type diabetes has been considered in this study.

3. Another issue is existing of an imbalanced-proportion of male and female (79.1% female). Furthermore, as the authors reported, compared with the general French population, NutriNet-Santé participants were younger (included of subjects > 15y), with higher educational and socio-professional levels and healthier dietary habits. They also had higher consumption levels of fruit and vegetables and lower levels of processed meats consumption. These may result in a bias in estimation of both exposure and outcome. As indicated by the authors (P12, lines 328-330), the characteristics of the study population might have led to an underestimation of associations with nitrites/nitrates from food additives, and an overestimation of those with nitrites and nitrates from natural sources.

4. A practical approach to response to some of the aforementioned concerns may be excluding younger adults (<30 y) from the analyses, and also conducting sensitivity analyses by separating male and female (use of sex-specific tertiles of exposure that were considered in categorizing nitrate/nitrite levels could not per se eliminate the concern). 

5. The authors considered both food products and water as naturally occurring sources of nitrite (or non-additive sources); I have concern about such categorization, because nitrite is considered as a contaminant agent in drinking water that is originated from fertilizers through run-off water, sewage, and mineral deposits. Labeling that nitrite, therefore, as natural source-originated may be misleading. A good suggestion may be separating sources of nitrite as foods-originated, water-originated and additives-originated, however, the estimated nitrite intake from drinking water was negligible (< 0.01% compared to 95.3% of total nitrite intakes from food sources). 

6. Although a main strength of the study is using a web-based self-administered 24h-dietary records (repeated at 6-month intervals) with a valid and reliable estimations of dietary intakes (validated by face-to-face dietician interview, and biological biomarkers), under-reporting of energy intakes (about 17%, as reported in the study flowchart) is questionable. 

7. The authors claimed that they used the French official regional sanitary control of tap water to estimate intakes of nitrite from drinking water. They need to provide more details regarding individual estimation of water-originated nitrite.

8. In the introduction section, authors stated that "In the mouth, much of the ingested exogenous nitrate is reduced to nitrite by commercial bacteria…."; this sentence need to be corrected as of ingested nitrate, ~25 % is extracted by the salivary glands and of which ~20 % is converted nitrite. It means that about 5 % of ingested nitrate is converted to nitrite in the oral cavity. This nitrite then is converted to NO in the stomach providing a continuous source of NO for human body. (see PMID: 35391922 for more details).

9. In the introduction authors state that eNOS is responsible for endogenous nitrate regulation; although eNOS is considered as the main source of NO production, which is converted to nitrite and then nitrate, the sentence is not factually correct that eNOS regulated nitrate.

10. Authors need to compare nitrate/nitrite intakes in this population with those from other populations (see for example, Babateen AM, Fornelli G, Donini LM, Mathers JC, Siervo M. Assessment of dietary nitrate intake in humans: a systematic review. Am J Clin Nutr. 2018;108:878-88). 

11. Page 8, line 213, authors state that date of an incident type-1 diabetes was considered for person time calculation, please provide data on incidence of T1D in the population. 

Minor comments

1. In the most Figures presented in the supplementary material, Y-axis is not started from zero. I strongly suggested that in all cases the start oy Y-axis to be zero. 

2. Page 10, lines 272-273: "As regards nitrites from food additives, the curve showed a logarithmic shape 273 with a plateau in the highly exposed participants (p for non-linearity =0.0009)". I did not find a figure in supplementation with logarithmic scales. Please add such a figure. 

3. I suggest that authors measure urine nitrate and nitrate in sub-sample to provide better indication of exposure to nitrate and nitrite. 

Reviewer #3: The authors investigated the possible associations between nitrite/nitrate intakes and their sources with development of type 2 diabetes in the framework of the French NutriNet-Santé cohort study (2009-2021). They reported an estimated population-mean intake of 194 and 5 mg/d for nitrate and nitrite, respectively and concluded that higher intake of nitrite from both natural and food additive sources increased risk of type 2 diabetes. Furthermore, they reported that there was no interaction between nitrate/nitrite intakes and antioxidants in relation to the development of diabetes, unlike those previously reported in a population-based cohort for nitrite and vitamin C. 

One of the main strength of the study was use of a national database for estimation of nitrate and nitrites, and assessment of details of commercial names/brand of industrial products to estimate individual additive exposure provides a strong support for estimation of nitrate and nitrites in this population. Some issues, however, should be considered for potential further evaluation: 

1. A low incidence rate of diabetes (~0.09%) during the study follow-up, an imbalanced-proportion of male and female in the study (79.1% female), including adolescents in the study participants, and self-reporting of the main outcome which seems resulted in an underestimation of the outcome, may decrease validity and reliability of the study findings. 

2. The authors discussed that the study participants had higher educational and socio-professional levels and healthier dietary habits (e.g., higher consumption levels of fruit and vegetables and lower levels of processed meats consumption); these characteristics may seriously affect estimation of nitrites/nitrates intakes from food additives that was reported about 4.7% of total intakes. 

3. The other confounders such as hypertension, history of CVD, TG/HDL-C and FPG level did not consider in their data analysis.

4. I do not agree with the authors regarding their definition about naturally occurring sources of nitrite (food products + water); nitrite is a contaminant agent in drinking water and categorizing it as a natural-originated compound is not acceptable. Moreover, I have some concerns about defining sources of nitrate/nitrite at all; as illustrated in Appendix 4a (Natural food (solid) sources of nitrites and nitrates) and Appendix 4b (Food sources of nitrites as food additives), "processed meats" were considered in both categories. Please be clearer about defining sources of nitrite and nitrate. 

5. The authors should also examine the association between dietary exposure to nitrites/nitrates with incident prediabetes or metabolic syndrome. 

6. The authors need to make a comparison between their estimations about dietary intakes of nitrate/nitrite with other populations (especially European populations) and probably previous estimations from their country. 

7. Regardless of using a validated method for estimating dietary intakes, the percent of the study participants that were excluded from the study due to under-report of energy intakes (about 17%), is considerable. Please discuss the issue and provide potential reasons, and address whether this missing information may affect the study generalizability. I strongly suggest that the authors make a comparison between under-reporters and other participant. 

Reviewer #4: See attachment.

Reviewer #5: Since the health effect of nitrites and nitrates on incident type-2 diabetes (T2D) is inconclusive and epidemiological and clinical evidence are lacking, this study aimed to examine the association between nitrites/nitrates and T2D through a population based prospective cohort study. The study also distinguished the effects of natural and food additive sources of nitrites and nitrates on T2D. Overall, I think this is a carefully designed and conducted cohort study and the research topic is of importance to population health. However, I am having a little concern on the web-based cohort and the self-administered questionnaire as I think the cohort may not be well representative of the population. Particularly, I think the study population under-represents the old people and people with low or no education. Below please find my specific comments. 

1. Competing Interests: please make sure that the funding from the IFIP does not have conflict of interest with this study.

2. Data Availability: noted that data are not fully publicly available. 

3. Line 139-140: given that the cohort is web-based and only participants with access to internet are recruited, is there any concern regarding the representativeness of the cohort. Since young people tend to use internet more than the old people, is there skewed age distribution in the recruited cohort? 

4. Line 156-158: since the surveys are all self-administered, I am worried whether the cohort will be representative of people with low or no education? Also, is there any concern on the quality of the self-administered survey? 

5. Line 195-198: I don't feel comfortable to ascertain people's T2D status through reimbursement of medication. First of all, if people do not want to disclose their disease status, is it ethical to track their medication reimbursement records to expose their disease status? Second, I am hesitant whether this can accurately ascertain people's T2D onset.. 

6. Line 222-224: when studying sodium nitrite, will adjusting for potassium nitrite (e249), natural nitrite, and overall nitrate intakes blind/obscure the effects of sodium nitrite?

[LINK]

---

## [Decision Letter · Decision Letter 2]

4 Nov 2022

Dear Dr. Srour,

Thank you very much for re-submitting your manuscript "Dietary exposure to nitrites and nitrates in association with type-2 diabetes risk:  Results from the NutriNet-Santé population-based cohort study" (PMEDICINE-D-22-02217R2) for review by PLOS Medicine.

I have discussed the paper with my colleagues and the academic editor and it was also seen again by all reviewers. I am pleased to say that provided the remaining editorial and production issues are dealt with we are planning to accept the paper for publication in the journal.

[LINK]

We hope to receive your revised manuscript within 2 weeks. Please email us (plosmedicine@plos.org) if you have any questions or concerns.

We look forward to receiving the revised manuscript by Nov 18 2022 11:59PM.   

Sincerely,

Callam Davidson, 

Associate Editor 

PLOS Medicine

plosmedicine.org

Comments from the Academic Editor:

As the reviewers noted, one of the limitations in this study is no information on the validity of their exposure assessment (nitrates, nitrites).

The authors' approach to this limitation in writing is substandard. They argued as follows. First, it would be ideal to have objective biomarkers for exposures to nitrates and nitrites. Second, there is no promising biomarker for those.

These two arguments partly support why the authors could not assess the validity of their exposure assessment. However, the arguments do not mean we can ignore the specific limitation. The authors should provide an interpretation and conclusion with the weakness into consideration. In other words, the authors should consider the possibility that the finding was an artefact for example.

Any readers would understand that the authors did great for their exposure assessment, including using multiple databases and collecting detailed dietary data. However, the objective view on no information on the validity (sensitivity and specificity) is crucial.

In the revised document, the authors should do the followings:

The authors should revise the abstract and state that they gained no information on the validity and that measurement errors would be substantial to question the positive association. Of note, the error characteristics are unclear, and the authors should not consider the bias-toward-the-null argument as the direction is unclear overall.

The conclusion sentences should be revised to those in the past tense (e.g. is to was) so that the authors can avoid generalizing the study finding to the general fact.

The authors should reduce their discussion on generalisability. The limitations of residual confounding and measurement errors mean that the result itself may be biased within the population. When the observation is biased, generalisability is the secondary concern.

In the method section, the authors should state no objective information on the validity measure.

In the discussion, the authors should clarify that validity, sensitivity, and specificity are unclear in their exposure assessment. The authors can keep the note about little availability of useful objective biomarkers, but should clarify that we must recognize the limitation.

In the discussion section, the authors may propose future validation study with a duplicate-meal design for multiple days sufficient to estimate habitual nitrate/nitrite intakes, for example. 

Biomarkers from tissues would be just one of many ways to assess the validity of dietary exposure.

split the long paragraph about strengths and limitations into two.

Minor revisions:

Revise Figure 1 and 2. Avoid showing the pie charts. Change each into a bar chat that has bars which lengths correspond to the average total amounts of the consumption and which internal segments show different sources. Pie charts are generally discouraged to display in a scientific article.

Figure 4. "estimation" should be "Point estimates". "Estimation" is a noun meaning what the authors conducted, not what the authors produced. The underbar "_" should be removed.

The x-axis of the spline figure should have a unit. The abbreviation requires an explanation in the legend.

Requests from Editors:

Line 80: Please update ‘made no difference’ to ‘did not distinguish’

The Data availability and Conflicts of interest sections can be removed from the main text – it is essential that all the relevant material is captured in the Data Availability Statement and Competing Interests sections of the submission form (this information will be published as metadata alongside your article).

Comments from Reviewers:

Reviewer #2: Regarding the revised-version of the manuscript entitled "Dietary exposure to nitrites and nitrates in association with type-2 diabetes risk: Results from the NutriNet-Santé population-based cohort study", although some of points have been addressed carefully, some issues need further clarifications. 

1- The authors have tried to discuss the important limitation of the current study, i.e., lack of representativeness of the study population, resulted in a lower incidence rate of T2D compared to French population (1% compared to 6%), high-estimated nitrate-nitrite exposure from food sources, and low-estimated nitrate-nitrite exposure from additive sources. They ascertained that their findings could not be generalized to other populations, even French population as well, but their conclusion addresses to updating regulations regarding nitrites from food additives and lack of beneficial effect of dietary nitrate from natural sources. I think that the conclusion should be limited to their study population (young women, with healthy lifestyle, higher socio-professional and educational levels that have mostly home access to the Internet).

2- Regarding T2D diagnosis, the authors reported that "Among the participants who provided blood sample during the clinical/biological examination, 232 had elevated fasting blood glucose (i.e. >1.26 g/L). Among them, 85.3% had consistently declared a T2D diagnosis and/or medication. Elevated blood glucose alone (i.e., without any declaration of T2D diagnosis or treatment) was not considered specific enough to classify the participant as a T2D case". It is not clear why they excluded about 15% of subjects with elevated blood glucose while according to IDF recommendation, fasting glucose higher than 126 mg/dL is a criteria for diagnosis of T2D. 

3- The results provided in response to reviewer 3 (comment 5) regarding possible association between nitrate-nitrite and risk of metabolic syndrome (i.e., significant association between total and food-originated nitrite with metabolic syndrome and lack of association with additive-originated nitrite) are in contrast with findings that provided for nitrite and T2D; how the authors explain this?

4- As pointed with reviewer 3 and 4, the estimation of dietary nitrate/nitrite needs to be validated using biological measurements. Although the authors have tried to discuss this important limitation, this issue remains challenging. 

5- In response to the comment 11, regarding the incident rate of T1D, the authors provided a rate of 12.3 per 100000 person-year, but they discussed that "Since the outcome of interest was T2D for this study, these incident T1D cases were censored at the date of diagnosis". This sentence is not make sense; please clarify. 

6- The response to minor comment 1 "In the most Figures presented in the supplementary material, Y-axis is not started from zero. I strongly suggested that in all cases the start of Y-axis to be zero" is not acceptable. Authors can use two-segment Y-axis to overcome this challenge. It can be found for example in the GraphPad Prism software. 

7- Please changes the following sentence: 

Original: In the mouth, about 5% of the ingested exogenous nitrate is reduced to nitrite by commensal bacteria residing in that area, and is then converted to nitric-oxide (NO) in the stomach, through the nitrate-nitrite-NO pathway, providing a continuous source of NO for human body, in addition to NO production through the L-arginine oxidative pathway.

Suggestion: About 25% of the ingested exogenous nitrate is reduced to nitrite by commensal bacteria residing in the mouth, of which about 20% is converted to nitric-oxide (NO) in the stomach, through the nitrate-nitrite-NO pathway (i.e., about 5% of ingested nitrate is converted to NO), providing a continuous source of NO for human body, in addition to NO production through the L-arginine oxidative pathway.

Reviewer #3: I have read the revised version of the manuscript entitled "Dietary exposure to nitrites and nitrates in association with type-2 diabetes risk: Results from the NutriNet-Santé population-based cohort study". The manuscript has been revised carefully according to the reviewers' comments, however, I have some points provided below: 

1- In response to comment 1 & 2, the lack of representativeness of the study population have been discussed in the revised version, however, the conclusion remained unchanged. The authors could not extrapolate their results to other populations.

2- In response to comment 5, new analyses were performed to assess the possible associations between nitrate/nitrite and risk of metabolic syndrome, and the results indicated a significant association between total and food-originated nitrite with metabolic syndrome and lack of association with additive-originated nitrite. These findings differed from those reported for T2D; would you please explain the new findings? 

Reviewer #4: Minor comment: Please correct 'Green leafy vegetables are the major source of dietary nitrates. Even though their role in reducing T2D risk has been suggested [1,2], the most recent meta-analyses [3-5] have reported low to very low quality of evidence for the association of incidence of cruciferous and green leafy vegetables with T2D risk' to 'Green leafy vegetables are the major source of dietary nitrates. Even though their role in reducing T2D risk has been suggested [1,2], the most recent meta-analyses [3-5] have reported low to very low quality of evidence for the association of cruciferous and green leafy vegetables with T2D risk'.

Reviewer #5: Thanks for the detailed responses to my comments/concerns. I think the authors have adequately addressed my and other reviewers' comments. I am happy with the revised version and would greenlight the paper for publication. Congratulations!

[LINK]

---

## [Editor Report · Decision Letter 3]

24 Nov 2022

Dear Dr Srour, 

On behalf of my colleagues and the Academic Editor, Dr Fumiaki Imamura, I am pleased to inform you that we have agreed to publish your manuscript "Dietary exposure to nitrites and nitrates in association with type-2 diabetes risk:  Results from the NutriNet-Santé population-based cohort study" (PMEDICINE-D-22-02217R3) in PLOS Medicine.

Please also make the following changes:

* Figure 2 legend should include 'and nitrates'

* Author Summary - Please add an additional single sentence bullet under 'What do these findings mean?' which summarises the study's main limitations.

PRESS

Sincerely, 

Callam Davidson 

Associate Editor 

PLOS Medicine